# Vitamin B6 deficiency disrupts serotonin signaling in pancreatic islets and induces gestational diabetes in mice

Ashley M. Fields [1], Kevin Welle[2], Elaine S. Ho [3], Clementina Mesaros[3] & Martha Susiarjo [1✉]

In pancreatic islets, catabolism of tryptophan into serotonin and serotonin receptor 2B (HTR2B) activation is crucial for β-cell proliferation and maternal glucose regulation during pregnancy. Factors that reduce serotonin synthesis and perturb HTR2B signaling are associated with decreased β-cell number, impaired insulin secretion, and gestational glucose intolerance in mice. Albeit the tryptophan-serotonin pathway is dependent on vitamin B6 bioavailability, how vitamin B6 deficiency impacts β-cell proliferation during pregnancy has not been investigated. In this study, we created a vitamin B6 deficient mouse model and investigated how gestational deficiency influences maternal glucose tolerance. Our studies show that gestational vitamin B6 deficiency decreases serotonin levels in maternal pancreatic islets and reduces β-cell proliferation in an HTR2B-dependent manner. These changes were associated with glucose intolerance and insulin resistance, however insulin secretion remained intact. Our findings suggest that vitamin B6 deficiency-induced gestational glucose intolerance involves additional mechanisms that are complex and insulin independent.

[1] Department of Environmental Medicine, University of Rochester School of Medicine and Dentistry, Rochester, NY, USA. [2] Mass Spectrometry Resource Laboratory, University of Rochester, Rochester, NY, USA. [3] Department of Systems Pharmacology and Translational Therapeutics, University of Pennsylvania Perelman School of Medicine, Philadelphia, PA, USA. ✉email: martha_susiarjo@urmc.rochester.edu

Normal pregnancy is associated with increased maternal glucose production and decreased insulin sensitivity to accommodate the energy need of the fetus; as such, pregnancy is associated with a persistent state of maternal insulin resistance[1,2]. The metabolic adaptations during pregnancy call for an increased demand for maternal insulin, which places stress on the pancreatic β-cells. To secrete sufficient levels of insulin during pregnancy, β-cells proliferate and increase in mass to fulfill the expanding maternal insulin demand and to prevent hyperglycemia[2–6]. Failed β-cell proliferation has been linked to maternal insulin insufficiency and hyperglycemia, and subsequently to the development of gestational diabetes in mice[7–9].

In 2010, German and coworkers reported that tryptophan-serotonin catabolism, a vitamin B6-dependent pathway, was highly upregulated in the pancreatic islets during pregnancy[9–11]. Dietary tryptophan restriction and genetic manipulation to this pathway reduced maternal islet serotonin levels and decreased maternal β-cell proliferation, ultimately leading to gestational diabetes in mice[9]. The role of vitamin B6 deficiency, however, has not been investigated. Although studies suggest that vitamin B6 deficiency in humans increases the risk for gestational glucose intolerance[12–14] and that vitamin B6 supplementation improves glucose tolerance during pregnancy[13,15], the causative link between vitamin B6 status during pregnancy and maternal glucose homeostasis, as well as the role of tryptophan-serotonin catabolism in vitamin B6 deficiency-associated state of glucose tolerance, has not been experimentally validated.

In this study, we hypothesized that maternal vitamin B6 deficiency perturbs the catabolism of tryptophan in the pancreatic islets and induces gestational diabetes. We investigated how gestational vitamin B6 deficiency influenced glucose homeostasis in pregnant mice and tested the impact of deficiency on serotonin levels in pancreatic islets. Results of this study provide new insights into the role of vitamin B6 on pancreatic islet physiology during pregnancy.

## Results

**Vitamin B6 deficiency induces glucose intolerance during pregnancy.** To determine whether vitamin B6 deficiency alters maternal glucose homeostasis, we induced vitamin B6 deficiency during pregnancy by feeding C57BL6/J female mice control or vitamin B6-deficient diet 3 weeks prior to pregnancy, and during mating and pregnancy. Liquid chromatography high-resolution mass spectrometry (LC-HRMS) analysis of maternal liver showed that pregnant mice from vitamin B6 deficiency group had decreased vitamin B6 level as measured by the increased ratio of kynurenine and kynurenic acid relative to controls at gestational day (GD) 12.5 and 16.5 ($p = 0.05$ and 0.0084, respectively; Supplemental Fig. 1a, b). Vitamin B6 deficiency did not significantly influence maternal body weight throughout gestation (Fig. 1a, Supplemental Table 1). In addition, we observed similar dietary intake between control and vitamin B6 deficiency dams throughout pregnancy (Supplemental Table 2).

To test whether vitamin B6 deficiency altered maternal glucose tolerance during pregnancy, we measured glucose levels after a 6 h fast and, subsequently, performed glucose tolerance testing (GTT) in pregnant mice at various gestational timepoints (i.e., GD 9.5, 12.5, and 16.5). At GD 9.5 fasting glucose levels were similar between treatment groups; at GD 12.5 and 16.5, however, maternal fasting glucose was elevated relative to controls (Fig. 1B). Analysis of glucose area under the curve (AUC) revealed that vitamin B6-deficient dams had higher total glucose levels compared to control at all gestational timepoints (i.e., GD 9.5, 12.5, and 16.5; Fig. 1F). Furthermore, glucose levels in vitamin B6-deficient dams were higher at $T = 15$ and 30 (GD 12.5;

Fig. 1D) and $T = 30$ (GD 16.5; Fig. 1E) while we did not observe specific timepoints that drove the increased glucose AUC in GD 9.5 dams (Fig. 1C). Combined analysis of fasting glucose and glucose time graphs suggest that the glucose intolerance in our vitamin B6-deficient mice starts around mid-gestation and worsens at the latter timepoints.

Gestational diabetes is defined as the onset of transient diabetes during pregnancy, thus, it is a pregnancy-specific condition[16]. To determine if the effects of vitamin B6 deficiency on maternal fasting glucose levels and glucose tolerance were pregnancy-specific, we performed similar tests in mice prior to pregnancy (i.e., non-pregnant) and at 3 weeks after delivery (i.e., postpartum). We found that vitamin B6 deficiency did not impact fasting glucose (Fig. 1B) or glucose tolerance (Fig. 1F) in non-pregnant or postpartum mice, and the glucose time course graphs were similar between the control and vitamin B6-deficient groups (Supplemental Fig. 2a, b). In addition, we did not observe the effects of vitamin B6-deficient diet on body weight and food consumption (Supplemental Tables 1 and 2, respectively) in non-pregnant mice. Our studies demonstrate that vitamin B6 deficiency-induced hyperglycemia and glucose intolerance are pregnancy-specific.

**Vitamin B6 deficiency is associated with impaired insulin sensitivity.** Glucose intolerance typically arises due to β-cell dysfunction[17–19] or peripheral insulin resistance[20]. To uncover the underlying mechanism of vitamin B6 deficiency-induced glucose intolerance, we performed in vivo glucose-stimulated insulin secretion (GSIS). Our data revealed that fasting insulin levels and insulin secretion were similar between control and vitamin B6-deficient dams at GD 12.5 (Fig. 2A) and 16.5 (Fig. 2C), suggesting that vitamin B6 deficiency did not impair insulin secretion. In fact, GD 12.5 dams had a significantly steeper slope (Fig. 2A), suggesting that more insulin was secreted over time relative to controls.

We next tested whether insulin resistance was the underpinning mechanism of vitamin B6 deficiency-induced glucose intolerance by performing insulin tolerance testing (ITT) in GD 12.5 and 16.5 dams. We observed that the insulin bolus was less effective at reducing glucose levels in vitamin B6-deficient dam compared to controls at GD 12.5 ($p = 0.0459$; Fig. 2B). At GD 16.5, however, vitamin B6-deficient dams had a normal, if not better, response, to the insulin challenge compared to controls at 30 and 120 min post insulin administration ($p = 0.0145$ and 0.0207, respectively; Fig. 2D). Overall, our results suggest that the vitamin B6-deficient dams are insulin resistant at GD 12.5, however, as pregnancy progresses, insulin sensitivity increases in vitamin B6-deficient dams.

**Vitamin B6 deficiency-induced glucose intolerance is associated with reduced maternal serotonin levels.** Michael German and colleagues demonstrated that reduced islet maternal serotonin during pregnancy is linked to gestational diabetes in mice[9]. Because vitamin B6 is a critical co-factor for serotonin synthesis (Supplemental Fig. 3), we asked whether its deficiency during pregnancy was linked to reduced islet-specific serotonin levels by using immunofluorescent co-staining of serotonin and insulin in GD 12.5 pancreases. Our studies revealed that serotonin co-localized with insulin, which defines pancreatic islets, but its staining was absent in the surrounding exocrine tissue; these results demonstrate that serotonin localization was specific to the islets (Fig. 3A). More importantly, serotonin immunostaining intensity in the pancreatic islets from vitamin B6-deficient dams was significantly reduced relative to controls ($p = 0.035$; Fig. 3B). To validate these observations, we used LC-HRMS to

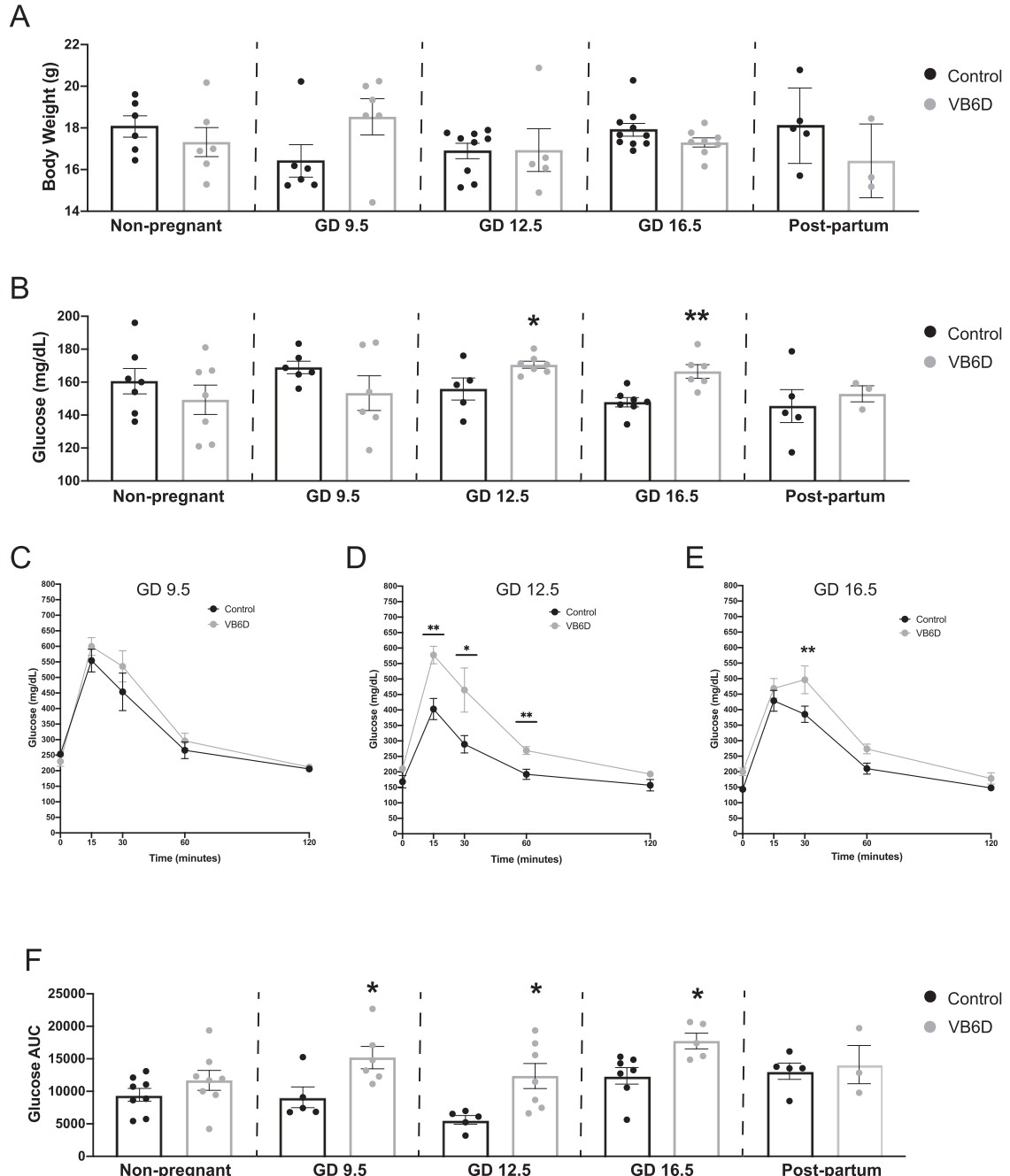

**Fig. 1 Vitamin B6 deficiency induces glucose intolerance during pregnancy.** Body weights (**A**) and fasting glucose levels (**B**) were measured in control and vitamin B6 deficient (VB6D) non-pregnant, pregnant, and 3 weeks postpartum mice. Glucose time graphs are shown for each timepoint GD 9.5 (**C**), GD 12.5 (**D**), and GD 16.5 (**E**) in which VB6D dams are glucose intolerant. Glucose area under the curve (AUC) was elevated in VB6D mice only during pregnancy (**F**). Data represent 3–8 mice per treatment group. Analysis of panels **A**, **B**, and **F** were done by an unpaired, two-sided, *t*-test. Panels **C**–**E** by two-way, repeated-measures ANOVA. In panels **D** and **E**, diet × time interaction effect was observed, and post hoc Tukey's tests were performed. *$p \leq 0.05$, **$p \leq 0.01$.

biochemically measure levels of metabolites in the tryptophan-serotonin catabolism pathway including serotonin, tryptophan, and pyridoxal 5′ phosphate (PLP) in pancreatic islets from control and vitamin B6-deficient GD 12.5 dams. Consistent with our immunostaining data, the LC-HRMS showed that vitamin B6-deficient dams had significantly decreased islet serotonin at GD 12.5 ($p = 0.0461$; Fig. 3C). In addition, tryptophan levels were significantly reduced in the pancreatic islets from vitamin B6-deficient dams ($p = 0.0480$; Supplemental Fig. 4a). Consistent

with our expectation, islets from GD 12.5 vitamin B6-deficient dams had significantly lower levels of PLP relative to controls ($p = 0.025$; Supplemental Fig. 4b).

Because our mouse model is a systemic dietary restriction of vitamin B6, we also investigated if the diet produced global decrease in maternal serotonin. Liquid chromatography–mass spectrometry (LC-MS) analysis revealed that maternal serum serotonin levels were lower in vitamin B6-deficient dams relative to controls at GD 16.5 ($p = 0.033$; Fig. 3E), but not at 12.5

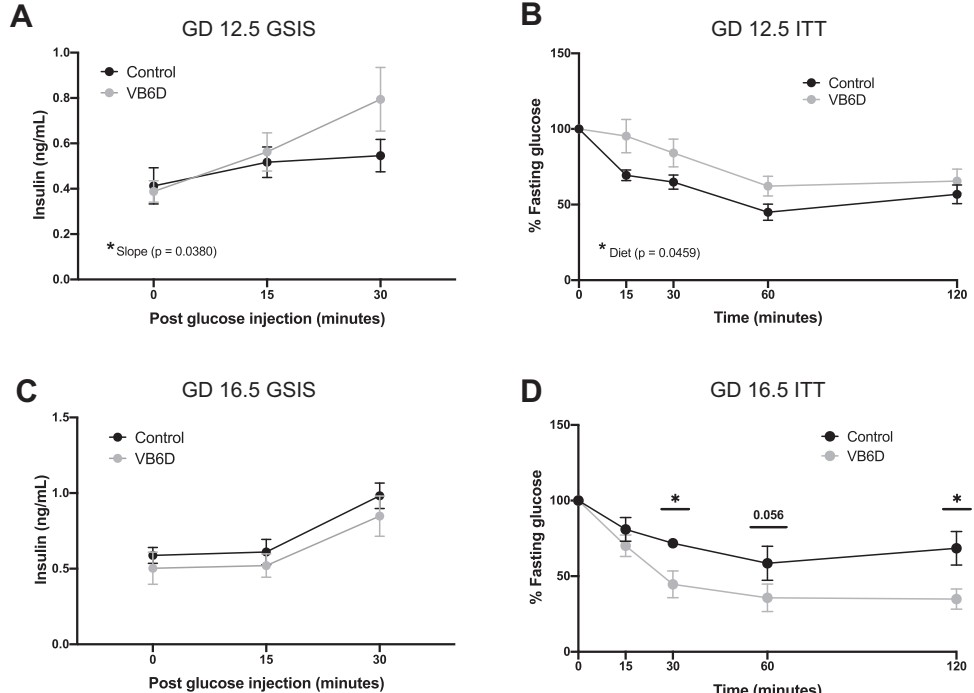

**Fig. 2 Vitamin B6-deficient dams have normal insulin secretion, but heightened insulin sensitivity during pregnancy.** At GD 12.5, VB6D dams have normal fasting insulin levels and insulin secretion response when glucose challenged compared to controls (**A**). At GD 12.5, VB6D has decreased insulin sensitivity compared to control (**B**). At GD 16.5, VB6D dams have normal fasting insulin levels and insulin secretion response when glucose challenged compared to control (**C**). At GD16.5, VB6D dams are more insulin sensitive compared to control indicated by a more significant reduction in glucose compared to controls (**D**). Data represent 5–8 mice per treatment group. Analysis of all panels was done by a two-way, repeated-measures ANOVA. A main effect of diet was observed in panel (**B**). In panel **D**, we identified a diet × time interaction effect, and a post hoc Holm–Sidak's test was subsequently conducted which revealed that the percent fasting glucose at $T = 15$ and 30 min were significantly different between the groups. $*p \leq 0.05$.

(Fig. 3D). These results suggest that islet-specific reduction of serotonin due to vitamin B6 deficiency occurs earlier than its systemic reduction in our mouse model.

**Vitamin B6-deficient pregnant mice have reduced β-cell proliferation.** Previous studies reveal that high levels of serotonin upregulate β-cell proliferation and increase β-cell mass during pregnancy[9]. To determine if reduced islet serotonin impaired β-cell proliferation and mass in our vitamin B6-deficient pregnant mice, we collected maternal pancreases at GD 12.5, which was the peak of β-cell proliferation during gestation[9], and subsequently performed immunofluorescence using the insulin and MCM-2 antibodies that marked proliferating β-cells. Despite having similar pancreatic wet weight (Supplemental Fig. 5a), vitamin B6-deficient mice had significantly reduced proliferating β-cells compared to controls ($4.831\% \pm 1.097$ vs $10.93\% \pm 0.7744$, respectively; $p = 0.0017$; VB6D + PBS vs Control + PBS panels in Fig. 4A and in the PBS group in Fig. 4B). Our results show that reduced islet serotonin in vitamin B6-deficient pregnant mice was linked to decreased maternal β-cell proliferation. Reduced β-cell proliferation in GD12.5 vitamin B6-deficient mice, however, was not associated with decreased β-cell mass as no significant difference was detected (Supplemental Fig. 5b), nor we observed changes in islet morphology (Supplemental Fig. 5c–e). As β-cell mass in mouse pregnancy reaches its highest ~4 days after the peak of proliferation[5], we collected pancreases from GD 16.5 mice and compared β-cell mass in vitamin B6-deficient and control groups. At GD 16.5, there was a trend in decreased β-cell mass ($p = 0.0726$) in vitamin B6-deficient dams relative to controls (Supplemental Fig. 5f).

**Vitamin B6 deficiency-induced glucose intolerance is serotonin receptor 2B (HTR2B) mediated.** Because our vitamin B6-deficient dams had decreased islet serotonin levels and reduced β-cell proliferation, we hypothesized that the vitamin B6 deficiency-induced glucose intolerance was mediated through disrupted serotonin signaling. Activation of the HTR2B, a G-protein coupled receptor that utilizes serotonin as a ligand[21,22], promotes β-cell proliferation during pregnancy[9]. Consistent with its function, *Htr2b* knock-out (KO) mice develop glucose intolerance only during pregnancy[9]. We hypothesized that the glucose intolerance in our vitamin B6-deficient mice was causatively associated with perturbed HTR2B signaling and that treatment with an HTR2B agonist would promote β-cell proliferation, ultimately improving glucose tolerance.

To determine whether treatment with an HTR2B agonist increases β-cell proliferation, we injected control and vitamin B6-deficient dams with BW-723C86 starting from GD 9.5 to 12.5, the critical window for *HTR2B*-dependent induction of β-cell proliferation[9], and quantified β-cell proliferation at GD 12.5. As expected, vitamin B6-deficient mice treated with the HTR2B agonist had significantly increased β-cell proliferation compared to the PBS-treated dams ($15.02\% \pm 0.3567$ vs $4.831\% \pm 1.097$, respectively, $p = {<}0.0001$, gray bars in Fig. 4B; VB6D + Agonist vs VB6D + PBS in Fig. 4A). Control dams treated with the agonist had a slight increase in β-cell proliferation compared to their respective PBS-treated controls ($13.31\% \pm 0.4738$ vs $10.93\% \pm 0.7744$, respectively, $p = 0.1771$, black bars in Fig. 4B; Control + Agonist vs Control + PBS in Fig. 4A), however, this was not statistically significant.

We next investigated whether the increased β-cell proliferation in the HTR2B agonist-treated vitamin B6-deficient mice

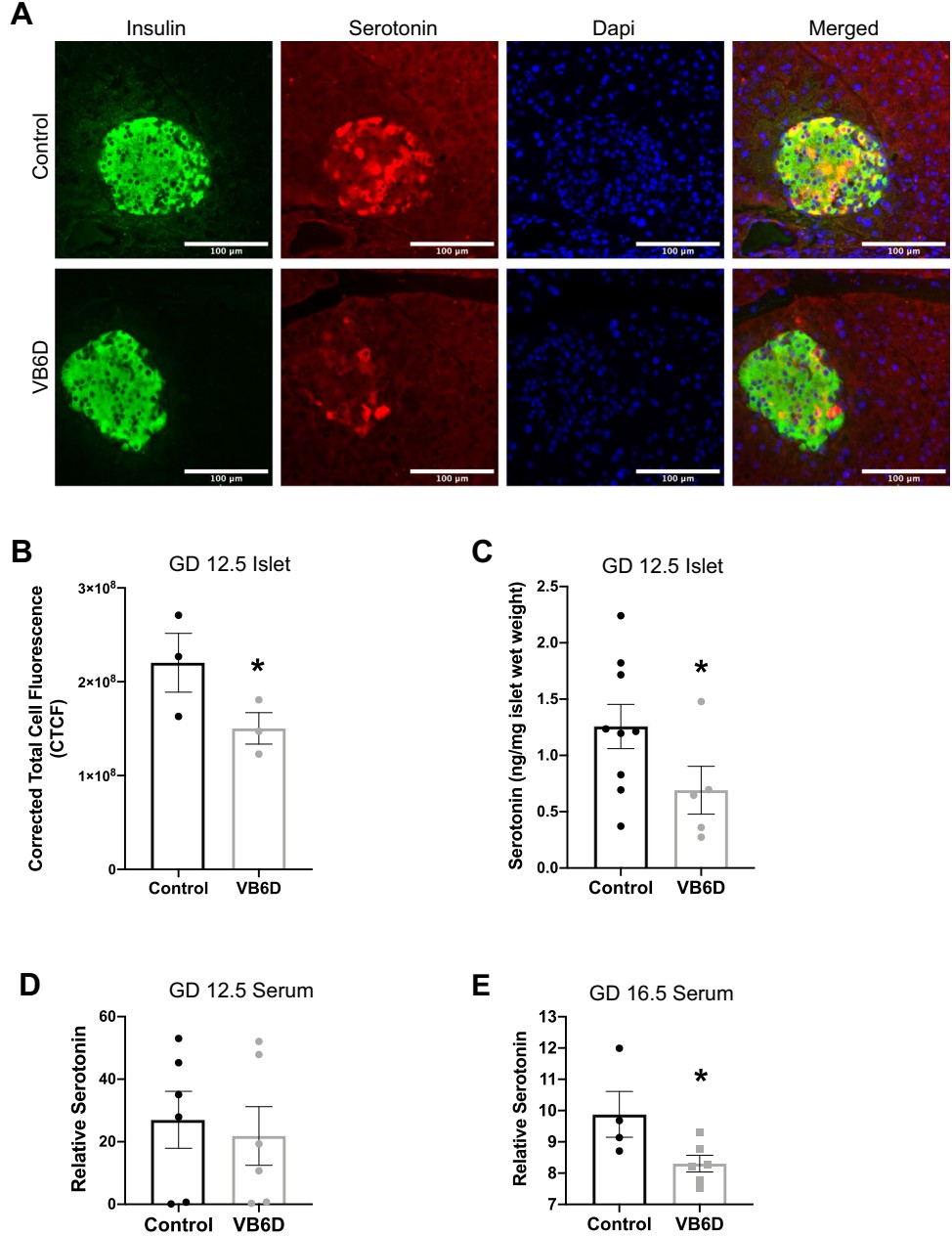

**Fig. 3 Reduced islet and serum serotonin in vitamin B6-deficient dams at GD 12.5 and 16.5, respectively.** Representative ×20 immunofluorescent images of GD 12.5 pancreases from control and vitamin B6-deficient or VB6D dams (**A**). Corrected total cell fluorescence (CTCF) was used to calculate islet serotonin in the GD 12.5 pancreases (**B**). In (**C**), we used LC-HRMS to measure absolute islet serotonin levels. Relative serum serotonin levels in GD 12.5 (**D**) and 16.5 (**E**) control and VB6D dams were measured using LC-MS. We analyzed 40–50 islets per mouse ($N = 3$ mice) in panels (**A**) and (**B**), 6–10 mice in panel (**C**), and 5–7 mice in panels (**D**) and (**E**). Unpaired, one-sided, $t$-tests were performed in (**B–E**). *$p \leq 0.05$; non-significant (ns) represents $p$ values > 0.1.

improved glucose tolerance. We injected control and vitamin B6-deficient dams with either PBS or BW-723C86 from GD 9.5 to 12.5 and performed GTT at GD 12.5 or 4 days later at 16.5. Our data revealed that the HTR2B agonist treatment did not influence GD 12.5 and 16.5 maternal weight (Supplemental Figs. 6a and 7a) or fasting glucose (Supplemental Figs. 6b and 7b) in control and vitamin B6-deficient dams compared to the PBS treatment groups. Consistent with our previous observation that vitamin B6 deficiency-induced glucose intolerance during pregnancy (Fig. 1F), in the PBS group, vitamin B6-deficient dams had significantly increased glucose AUC relative to controls at GD 12.5 ($p = 0.0186$; Supplemental Fig. 6d) and 16.5 ($p = 0.0313$; Fig. 4C). Upon HTR2B agonist treatment,

GD12.5 vitamin B6-deficient dams remained glucose intolerant compared to agonist-treated controls dams ($p = 0.0290$; the agonist group in Supplemental Fig. 6d and triangle symbol in Supplemental Fig. 6c). In contrast, at GD 16.5, the HTR2B agonist treatment ameliorated the glucose-intolerant phenotype in vitamin B6-deficient mice compared to PBS-treated vitamin B6-deficient dams ($p = 0.0185$; gray bars in Fig. 4C and gray time graphs in Supplemental Fig. 7c). Collectively, our results demonstrate that vitamin B6 deficiency-induced glucose intolerance is mediated through HTR2B-dependent disruption of serotonin signaling, and intervention with an HTR2B agonist restores β-cell proliferation at GD 12.5, but the improvement in glucose tolerance is not observed until GD 16.5.

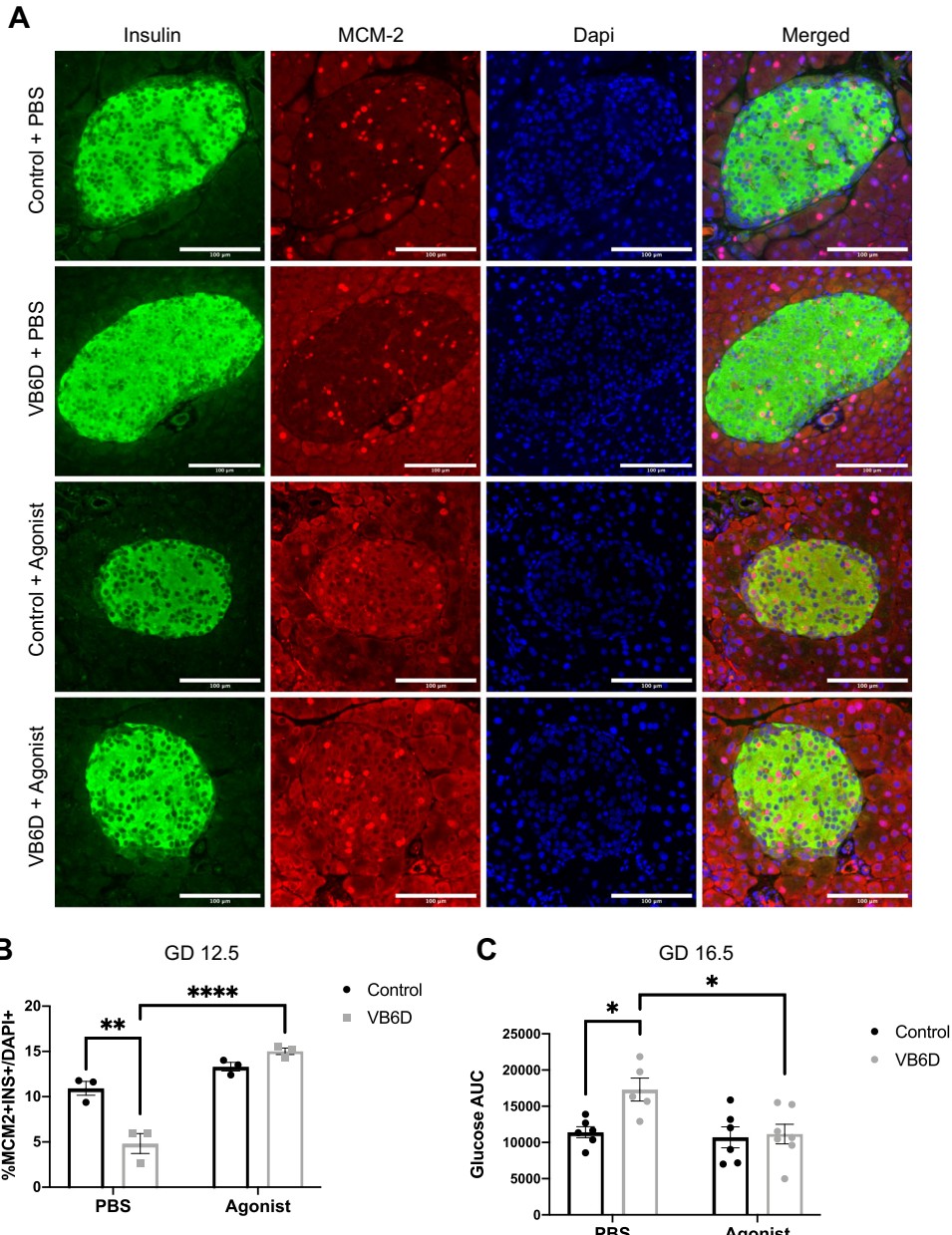

**Fig. 4 HTR2B receptor agonist treatment increases β-cell proliferation in vitamin B6-deficient dams and normalizes glucose tolerance at GD 16.5.**
Shown in (**A**) is representative ×20 immunofluorescent images of GD 12.5 pancreases from PBS and HTR2B agonist injected control and vitamin B6-deficient (VB6D) mice. Proliferating β-cells are identified using MCM-2 and Insulin antibodies and calculated as percentage of total β-cells (**B**). Glucose AUC for PBS and HTR2B agonist-treated control and VB6D dams at GD 16.5 is shown in (**C**). Data for panels **A** and **B** are based on 40–50 islets per mouse ($N = 3$ mice), while panel **C** represents 5–7 mice. Two-way ANOVA tests performed in panels **B** and **C** revealed a diet × treatment interaction effect and post hoc Tukey's multiple comparison tests were subsequently performed. *$p \leq 0.05$, **$p < 0.01$, ****$p < 0.0001$.

## Discussion

The goal of the current study is to elucidate a previously unknown role of vitamin B6 in the tryptophan-serotonin catabolism pathway in the pancreatic islets, specifically during β-cell proliferation in pregnancy. Our laboratory is the first to experimentally demonstrate that vitamin B6 deficiency during pregnancy is associated with hyperglycemia and glucose intolerance. Although pregnancy itself induces a "physiological state of glucose intolerance"[23,24], our studies show that vitamin B6 deficiency exacerbates gestational glucose intolerance. Mechanistically, these metabolic phenotypes are associated with reduced serotonin levels in maternal pancreatic islets and decreased β-cell proliferation. Interestingly, treating vitamin B6-deficient dams

with an HTR2B agonist increased β-cell proliferation and improved gestational glucose intolerance, suggesting that vitamin B6-induced glucose intolerance is causatively linked to HTR2B activation. Previous work demonstrated a role for HTR2B-mediated serotonin signaling in regulating maternal glucose homeostasis in mice[9]. Although pharmacological inhibition of the HTR2B receptor has been previously shown to induce gestational diabetes in pregnant mice[9], our study is the first to use an HTR2B agonist to increase β-cell proliferation and improve glucose tolerance during pregnancy. Furthermore, our results show that islet serotonin levels are decreased while the peripheral serotonin remains unaltered at GD 12.5 of pregnancy. Decreased islet serotonin was associated with reduced levels of PLP, the

essential co-factor for its synthesis. Although we did not measure expression and activities of tryptophan hydroxylase 1 (TPH1), the first and rating-limiting enzyme in the tryptophan-serotonin catabolism pathway, our LC-HRMS metabolite analyses revealed that tryptophan was catabolized properly, if not higher, in the islets of the vitamin B6-deficient mice (Supplemental Fig. 4a), suggesting that the enzymatic activity of TPH1 was not impaired. An interpretation of these results is that the reduced islet serotonin originates downstream of the TPH1 activity, for example from alteration in the PLP-dependent conversion of 5-hydroxytryptophan into serotonin (Supplemental Fig. 3). In sum, our findings are consistent with previous studies showing that modulators of tryptophan-serotonin catabolism influence an individual's risk of developing gestational diabetes[1,25] and provides novel experimental evidence that vitamin B6 deficiency during pregnancy is a potential risk factor for gestational diabetes.

Glucose intolerance can arise from impaired β-cell insulin secretion[17–19] or insulin resistance[20]. Our studies revealed that the vitamin B6-deficient dams had normal β-cell insulin secretion (Fig. 2A, C), suggesting that the residual β-cells secrete sufficient levels of insulin although it may be less effective in clearing glucose from the blood stream. At GD 12.5, we observed our vitamin B6-deficient mice were insulin resistant; interestingly, at GD 16.5, however, the dams had a heightened insulin-mediated glucose response when given a relatively low dose (0.5 U/kg*body weight) of insulin (Fig. 2D). These observations suggest that the glucose intolerance in our vitamin B6-deficient mice is truly complex and that both insulin resistance and other insulin-independent mechanisms contribute to the phenotype. Potential insulin-independent mechanisms include alterations in glucose uptake due to differences in activities of the glucose transporters GLUT4[26] and GLUT1[27], as well as impaired "glucose effectiveness" which refers to the ability of glucose to promote self-uptake and suppress its own production[28–30]. Furthermore, the increased insulin sensitivity in the GD16.5 vitamin B6-deficient mice may indicate an abnormal brain-related glucose counterregulatory mechanism that is necessary for restoring euglycemia[31]. Future studies should focus on these additional mechanisms through detailed investigations on glucose transporters, glucose effectiveness by utilizing advanced methods such as intravenous GTT[32] and/or hyperglycemic clamp[33], as well as various counter-regulatory factors including glucagon, epinephrine, and cortisol[34].

Our work established that vitamin B6 deficiency induces hyperglycemia during pregnancy at GD 12.5 and 16.5 (Fig. 1B). Although our control dams have higher fasting glucose values compared to other studies[1,9,35,36], they are within the expected range reported after a 6 h fast in mice[37]. Factors that influence fasting glucose levels in mice include the measurement method (i.e., glucometer vs enzymatic assay[38]), type of glucometer (i.e., human vs animal[38,39]), fasting duration (i.e., 6 h vs overnight[38,40]), strain/substrain[37,38], and repeated stress[41,42]. PBS injected GD 12.5 and 16.5 control dams in the agonist studies (Supplemental Figs. 6b and 7b) indeed had elevated fasting glucose relative to non-injected controls (Fig. 1B) which may reflect daily injection-induced stress.

As part of this study, we created a novel mouse model of gestational diabetes through the generation of mice that are vitamin B6-deficient through dietary restriction of pyridoxine HCl. Although we have not metabolically characterized the long term, post-pregnancy impact of vitamin B6 deficiency in our mice, our mouse model provides complementary insights into the various mechanisms underlying gestational diabetes in humans. The other mouse models of gestational diabetes include the streptopzocin (STZ)-treated[43–46] and db−/+[47–49] mice which have both strengths and limitations. The STZ-treatment induces glucose intolerance through chemical destruction of β-cells[43–46]

which represents a distinct mechanism compared to our mouse model. Another difference is that the STZ-induced glucose intolerance is not specific to pregnancy[50]. The db−/+ model of gestational diabetes, on the other hand, has similarities to our model including glucose intolerance that develops mid-gestation[47–49], however, the phenotype is induced by genetic deletion of the leptin receptor, a different pathoetiology relative to our model. To our knowledge, our vitamin B6-deficient mouse model is one of the first nutritional mouse models of gestational diabetes that disrupts serotonin signaling needed for β-cell proliferation and leads to glucose intolerance in a partially HTR2B-dependent manner. As vitamin B6 deficiency is the leading nutrient deficiency in the United States, our mouse model provides a potentially useful tool to investigate novel mechanisms underlying gestational diabetes that are relevant to humans.

Although our vitamin B6 deficiency mice were generated with human-relevant levels in mind, it is important to note that the causes of vitamin B6 deficiency in developed countries such as the United States are unlikely to be solely dietary-derived, since most Westernized diets are enriched with vitamin B6[51,52]. Genetic variants in the ALPL gene, for example, can highly influence vitamin B6 status[53–55] as this enzyme converts vitamin B6 into a membrane soluble isoform, allowing this micronutrient to reach target tissues and cells[55,56]. Furthermore, levels of estrogens, derived from oral contraceptive[57,58] as well as endogenous sex hormone[59], could also influence vitamin B6 status suggesting that factors that modulate estrogen levels could play a role in impacting vitamin B6 status. The data presented in this current manuscript are proof of principle evidence showing that altered vitamin B6 status during pregnancy is causatively linked to gestational diabetes. Our work provides the rationale to further study how genetic and environmental factors that disrupt vitamin B6 bioavailability may impact maternal glucose homeostasis during pregnancy.

In conclusion, we have demonstrated that adequate levels of vitamin B6 during pregnancy support a healthy state of maternal glucose metabolism. Future studies should focus on under-standing how factors that can potentially perturb vitamin B6 bioavailability during pregnancy may modulate the tryptophan-serotonin pathway and impact maternal glucose homeostasis.

## Methods

**Mouse information.** Six-to-eight-week-old virgin female C57BL/6J (JAX) mice were fed either a control (6 mg/kg pyridoxine hydrochloride or HCl; TD. 150561; Envigo, Madison) or vitamin B6-deficient (0.5 mg/kg pyridoxine HCl; TD. 160738; Envigo, Madison) diet, 3 weeks prior to mating, during mating, and throughout pregnancy. Vitamin B6 content in our control diet was comparable to standard laboratory mouse chows[60]. To minimize potential confounding effects from phytoestrogens, the control and vitamin B6-deficient diets were corn- instead of soy-based. In addition, to ensure no additional vitamin B6 is present, the diets used vitamin-free casein. The assigned dietary treatments continued until the experimental endpoints (i.e., GD 9.5, 12.5, or 16.5) as stated below. Details on diet composition are provided in Supplemental Table 3. Mice were housed in XJ microisolator cages (Allentown) with 1/8″ pelleted cellulose performance bedding (BioFresh) in a 12:12-h light-dark cycle room (i.e., lights on 6:00 am; lights off 6:00 pm) at 74 °F. All mouse work was carried out with the approval of the University Committee on Animal Resources of the University of Rochester School of Medicine and Dentistry.

**Creation of the vitamin B6-deficient mouse model.** We designed the vitamin B6-deficient diet based on a previous study that characterized the association between varying levels of pyridoxine HCl (a common dietary isoform of vitamin B6) in the diet and the circulating plasma levels of pyridoxal 5′ phosphate or PLP (the bioactive and most common isoform in blood) in non-pregnant mice[61]. We extrapolated these data to design a diet with a concentration of pyridoxine HCl (i.e., 0.5 mg/kg) that would result in plasma level of <20 nmol/L PLP (Supplemental Fig. 8), the threshold for vitamin B6 deficiency in humans[62]. The vitamin B6-deficient diet used in the current study is predicted to produce plasma PLP levels of 18.5 nmol/L.

**Indirect measurement of vitamin B6 status**. Kynurenine aminotransferase, the enzyme that converts kynurenine to kynurenic acid (Supplemental Fig. 3), is vitamin B6-dependent, as such the ratio of kynurenine to kynurenic acid has been used as an indirect measurement of vitamin B6 status[63]. To determine if dams fed the vitamin B6-deficient diet had low vitamin B6 status, we used LC-HRMS to compare the relative levels of kynurenine and kynurenic acid in GD 12.5 and 16.5 maternal livers from control and vitamin B6-deficient dams.

Frozen livers were cut on a tile kept on dry ice to 10–30 mg aliquots and weighed on an analytical balance. Tissues were then added to 0.2 mL 80% cold PBS spiked with 1 μg of each internal standard ($[^{13}C_{11}]$-tryptophan and $[^{13}C_4]$-kynurenine, 20 μL of 50 ng/μL in water, both from Cambridge Isotope Laboratories). Subsequently, all samples were homogenized using a Bullet Blender 24AU (Model BB24-AU) for 5 min at power 12 at 4 °C. Following homogenization, the samples were incubated on ice for 20 min to allow for metabolites extraction from tissues into the methanol solution. The tubes were hand vortexed for 5 s each, then spun down for 5 min at 13,500 rpm and the clear supernatants were moved to a clean Eppendorf tube. Supernatants were dried under nitrogen and resuspended in 100 μL of water then transferred to HPLC vials and 2 μL injections were made. Calibration curves were constructed for all of the analytes using authentic standards and the same amounts of internal standards.

Kynurenine and kynurenic acid were analyzed by LC-HRMS as reported in ref. [64] with the only modification that the LC was coupled to a Q Exactive-HF with a heated ESI source operating in negative ion mode alternation full scan and MS/MS modes. The $[M-H]^-$ ion of each analyte and its internal standard was quantified, with peak confirmation by MS/MS.

**GTT**. Non-pregnant, pregnant (GD 9.5, 12.5, and 16.5), and 3-week postpartum mice were fasted for 6 h (7:00 am to 1:00 pm) and subsequently injected with 2 g/kg body weight of glucose intraperitoneally[40,65]. Prior to injection ($T = 0$) and 15, 30, 60, and 120 min post injection (i.e., $T = 15$, 30, 60, and 120, respectively), blood was sampled from the tail vein using an ALPHATRAK2 glucometer (AlphaTrak). To measure total glucose concentration for each mouse, we calculated the AUC of the glucose time course graph. Differences in AUC among the treatment groups were compared using the statistical analysis described below.

**GSIS**. GD 12.5 and 16.5 pregnant mice were fasted for 6 h (7:00 am to 1:00 pm) and subsequently injected with 2 g/kg body weight of glucose intraperitoneally[40,65]. At $T = 0$, 15, and 30, tail vein blood was collected using a heparinized capillary tube (ThermoFisher). Blood samples were expelled from the capillary tube into a microcentrifuge tube and placed on ice until end of experiment. The samples were centrifuged for 10 min at 8000 rcf at 4 °C and the plasma was transferred to a new tube and stored at −80 °C until insulin levels were measured. Plasma insulin was analyzed using an ultra-sensitive mouse Insulin ELISA kit (Crystal Chem), according to manufacturer's wide range protocol (0.1–12.8 ng/mL insulin) and the colorimetric endpoint was measured using the Epoch plate microplate spectro-photometer (BioTek) with Gen5 software (BioTek).

**ITT**. GD 12.5 and 16.5 pregnant mice were fasted for 6 h (7:00 am to 1:00 pm) and subsequently injected 0.5 U/kg body weight of Insulin (Humulin® U-100)[40] intraperitoneally. At $T = 0$, 15, 30, 60, and 120, blood was sampled from the tail vein using an ALPHATRAK2 glucometer (AlphaTrak). At each timepoint, data were represented and analyzed as a percentage of fasting glucose. Comparisons between treatment groups were performed, using the statistical analysis described below.

**HTR2B agonist injections**. An HTR2B agonist, BW-723C86 (Sigma-Aldrich), was dissolved in phosphate-buffered saline (PBS) to a concentration of 0.25 mg/mL[66]. Pregnant dams were subcutaneously injected daily with either 100 μL of BW-723C86 (1 mg/kg bw dose) or PBS vehicle from GD 9.5 to 12.5 at the time when *Htr2b* mRNA expression peaks[9].

**Pancreas immunohistochemistry (IHC)**. Pancreas extraction was performed using a published protocol[67]. Briefly, GD 12.5 and 16.5 pancreases were isolated, fixed by immersion in 10% neutral buffered formalin (Sigma-Aldrich) for 10 h at 4 °C, paraffin-embedded, and cut into 5-μm sections. Slides were deparaffinized in xylene and rehydrated to distilled water using graded ethanol. We performed heat-induced epitope retrieval with sodium citrate buffer (10 mM sodium citrate, 0.05% Tween-20, pH 6.0) in a 95 °C water bath for 20 min and slides were membrane permeabilized with 0.2% Triton X-100 in PBS for 10 min. For IHC-immuno-fluorescence, slides were incubated in 0.1 M glycine for 60 min to quench auto-fluorescence due to the fixation process. Prior to immunostaining, sections were blocked with 5% normal donkey serum (IHC-immunofluorescence) or 5% bovine serum albumins or BSA (IHC-chromogenic) for 60 min. Primary antibodies used were rabbit anti-insulin (1:200, Abcam; ab18547), mouse anti-MCM-2 (1:750, ThermoFisher; MA5-15895), and rat anti-serotonin (1:200, Abcam; ab6336). Secondary antibodies used were 1:200 donkey anti-Rabbit-AlexaFluor® 488 (Jackson-nImmuno), 1:750 donkey anti-Mouse-AlexaFluor® 647 (JacksonImmuno), and 1:200 donkey anti-Rat-AlexaFluor® 647 (JacksonImmuno). Primary antibodies were diluted with 1% BSA and 0.1% Tween-20 in PBS, and slides incubated

overnight at 4 °C. We washed the slides using Millipore water for 3 × 5 min and subsequently incubated the slides in secondary antibodies at room temperature in the dark for 2 h. Nuclei were stained with 300 nM DAPI (Invitrogen).

**Serotonin detection and measurement**. Serotonin was detected using immuno-fluorescence with the antibodies stated above. Multichannel images were acquired of Insulin (Alexa488), Serotonin (Alexa647), and DAPI using an Upright ECLIPSE Ni-E microscope (Nikon) at an ×20 objective. All images were captured with equal exposure times and laser intensities best determined for each channel. Using the insulin antibody, the pancreatic islets were located. The fluorescent serotonin staining intensity in each islet was evaluated by converting the images to gray scale in ImageJ (v1.48, NIH). Relative levels of serotonin immunostaining intensity were quantified using the corrected total cell fluorescence (CTCF) method[68]. Briefly, using measurement features in ImageJ, a region of interest was created by tracing the islets. Islet area and mean gray value were subsequently measured, and integrated density for the islet was calculated by multiplying area and mean gray value. In addition, background readings of three similar-sized areas, adjacent to the islet, were measured. Last, relative serotonin intensity was determined using the formula: CTCF = integrated density of traced islet − (area of traced islet × average of the mean gray value of background readings). An image representation can be found in the supplemental section (Supplemental Fig. 9).

**LC-MS analysis of serum serotonin**. On GD 12.5 and 16.5, control and vitamin B6-deficient pregnant dams were euthanized, blood serum collected via cardiac puncture, and stored at −80 °C until analyses. Twenty microliters of each serum sample was transferred to a new tube and $[^2H_4]$-serotonin (internal standard) was added to a final concentration of 100 ng/mL. Nine volumes (180 μL) of acet-onitrile were added to each sample to precipitate the proteins. Tubes were vortexed and then centrifuged at 16,000 × g for 5 min. The supernatant was collected and placed in an autosampler vial.

For serotonin analysis, 5 μL of each sample was injected using an Ultimate 3000 UHPLC (Dionex) onto a 2.1 × 100 mm Accucore HILIC column, with 2.6 μm beads (ThermoFisher). Analytes were eluted using a mobile phase gradient and analyzed in positive mode on a TSQ Quantum Access Max triple quadrople mass spectrometer (ThermoFisher) with a heated electrospray ionization source. The spray voltage was set to 4.5 kV, the sheath gas flow rate was set to 65, and the auxiliary gas flow rate set to 5, while the capillary temperature was set to 400 °C. The mobile phases were A: 20 mM ammonium formate in 0.1% formic acid, and B: 20 mM ammonium formate in 90% acetonitrile with 0.1% formic acid. The flow rate was set to 800 μL/min, and the column oven was set to 25 °C. The gradient began at 100% B and held for 0.75 min before ramping to 50% B in 0.5 min, where it was held for 0.5 min, and returned to 100% B in 0.25 min. The column was re-equilibrated at 100% B for 1 min prior to the next injection.

The precursor ion for serotonin was 177.1 *m/z*, with fragment ions of 115 and 160 *m/z* with collision energy of 28 and 11, respectively, at a tube lens voltage of 86 V. The precursor ion for serotonin d4 188.1 *m/z*, with fragment ions of 188.2 and 164.1 *m/z* with collision energies of 28 and 11 respective to a tube lens voltage of 65 V. Peak areas corresponding to levels of endogenous serotonin or the spiked internal serotonin standard were calculated using the LC Quan node of the XCalibur software (ThermoFisher). To determine the relative levels of serum serotonin, we calculated the ratio between endogenous serotonin and the internal serotonin standard.

**Pancreatic islet isolation**. Pancreatic islets were isolated using the histopaque gradient method[69]. Briefly, GD 12.5 control and vitamin B6-deficient dams were anesthetized using ketamine/xylazine mixture[70]. We digested the extracted pan-creas in Hank's buffer-saline solution (HBSS) with 2 mg/mL collagenase P for 15 min at 37 °C on an orbital shaker at 225 rpm. Immediately after digestion, the pancreatic tissues underwent a series of washes with ice-cold HBSS and, subse-quently, was filtered through a 380 μm stainless steel mesh filter into 10 mL of ice-cold HBSS to remove pancreatic exocrine tissue. The pancreatic islets were isolated from the filtrate by histopaque 1100 gradient centrifugation at room temperature. The histopaque supernatant, containing the pancreatic islets, was decanted into ice-cold PBS and washed three times with PBS. Pancreatic islets were pelleted, the supernatant was subsequently discarded, and the islets were weighed to obtain wet weight, and immediately stored at −80 °C until further use.

**LC-HRMS of metabolites in pancreatic islets**. After islets were extracted from GD 12.5 control and vitamin B6-deficient dams, 50 μL of an internal standard mix ($[^2H_4]$-serotonin, $[^{13}C_{11}]$-tryptophan, and $[^{13}C_4]$-kynurenine) were added to each sample and blank tubes. One milliliter of 80% ice-cold methanol was added to each tube. Samples were pulse sonicated in ice and incubated for 10 min to allow for metabolites extraction from tissues into the methanol solution. The tubes were vortexed for 10 s and spun down for 5 min at 13,000 rpm at room temperature. The clear supernatants were moved to a clean Eppendorf tube, dried under nitrogen, and resuspended in 100 μL of 0.1% formic acid, and subsequently transferred to HPLC vials.

Two microliters of each sample were injected onto a Hypersil Gold column (2.1 × 150 mm, 1.9 μm). A 15-min gradient using two phases A (0.1% formic acid)

and B (0.1% formic acid in methanol) was used for metabolites separation, starting from 100% A to 98% A at 6 min, 1% A at 10 min (held constant for 2 min), and at 12.1 min back to 100% A. The MS conditions were as described in ref. [71]. Calibration curves were constructed for all of the analytes using authentic standards and the same amount of internal standards as used for the islet samples. The area ratio of analyte to internal standard was plotted versus the amount of analyte and a linear regression was used to calculate the amount of metabolite in the tube. The calculated amount was adjusted to the wet weight of the islets.

For islet serotonin quantification, a seven-point calibration curve was constructed using the internal standard. The linear equation of the curve is $Y = 0.17*X + 0.28$ (Supplemental Fig. 10a). For semi-quantitative analysis of islet PLP, we used the kynurenine internal standard due to its similar expected retention time as PLP (i.e., 3.9 min for kynurenine vs 4.2 min for PLP). The linear equation for the kynurenine calibration curve is $Y = 0.878*X - 7.36$ (Supplemental Fig. 10b).

**Statistics and reproducibility**. Statistical analyses were performed using Prism 8 (Graphpad, San Diego, CA). Boxplots, Shapiro–Wilk's test, and the Levene's test were used to assess outliers, normality, and homogeneity of variances, respectively. An unpaired student's $t$-test, two-sided was conducted on the following data sets: body weight measurements, fasting glucose, glucose AUC, islet morphology, and islet tryptophan. An unpaired student's $t$-test, one-sided was conducted on the serum serotonin, and islet serotonin and PLP, and indirect vitamin B6 measurement data sets. Two-way, repeated measures, ANOVA with post hoc Holm–Sidak's tests were conducted on glucose and insulin time graphs, and GSIS data sets. Two-way ANOVA with post hoc Tukey's test was conducted on the β-cell proliferation and glucose AUC data from the PBS and HTR2B agonist-treated control and vitamin B6-deficient pregnant dams. All statistical tests were performed with an alpha = 0.05 and power = 0.80. $P$ values ≤0.05 were considered statistically significant. Data are represented as mean ± SEM throughout this manuscript except when stated otherwise.

**Reporting summary**. Further information on research design is available in the Nature Research Reporting Summary linked to this article.

## Data availability

Data that are reported in our manuscript will be maintained in the corresponding authors' backup server. Data details will be released by a reasonable request to the corresponding author. Source data underlying plots shown in figures are provided in Supplementary Data 1.

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

## Acknowledgements

The work was supported by the National Institute of Environment Health Sciences (RO1 ES029469-01A1 and P30 ES001247 to M.S.; T32 ES007026 to A.M.F.; P30ES013508 to C.M.) and an instrumentational grant from the National Institute of Health (S10 OD025242 to K.W.). We acknowledge Kavaljit H. Chhabra at University Rochester Medical Center for his technical support in our ITT and GSIS studies and helpful discussion on the manuscript.

## Author contributions

A.M.F. contributed to the writing of the manuscript, performing the majority of the experiments, and analyzing the data. K.W. contributed to the writing, designing, and performing the method of liquid chromatography-mass spectrometry for serum serotonin. E.S.H. and C.M. contributed to the writing, designing, and performing the high-resolution mass spectrometry studies. M.S. conceived the original ideas, supervised the studies, interpreted data, and contributed to the writing of the manuscript.

## Competing interests

The authors declare no competing interests.

## Additional information

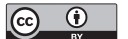

