## [Peer Review File · Communications Biology]

Reviewers' comments:

Reviewer #1 (Remarks to the Author):

Fields et al examine the role of vitamin B6 deficiency in the development of gestational diabetes mellitus (GDM). The authors use a vitamin B6 deficient diet to induce vitamin B6 deficiency in pregnant mice. Using this approach, the authors investigate whether vitamin B6 deficiency decreases serotonin levels in pregnant mice and whether this affects the adaptive proliferative response of beta cells during pregnancy to alter maternal glycemic control. The authors report that vitamin B6 deficiency decreased serotonin levels and beta cells proliferation in pancreatic islets from pregnant mice. These changes were associated with impaired glucose tolerance in pregnancy. The experiments are straightforward and while the data are interesting, they provide minimal mechanistic insight. Moreover, further characterization of the metabolic phenotype of the mice during pregnancy is necessary.

Specific Comments:

1. Did vitamin B6 deficiency affect body weight gestational weight gain or food intake in the mice? Tables should be provided in the paper.
2. The authors claim they are modeling gestational diabetes, but vitamin B6 deficiency is present throughout pregnancy. A more detailed characterization of maternal glycemia at earlier timepoints in pregnancy would be appreciated.
3. The authors do not show that there are impacts of vitamin B6 deficiency on insulin. How did vitamin B6 deficiency affect circulating insulin and insulin following glucose administration (on the GTT)? Functional assays of glucose stimulated insulin secretion by the islets are also necessary to determine the function impact on the islet.
4. Beyond the histological analyses reported in this study, the analyses should be supported by biochemical measures of insulin and serotonin within the mouse islet.
5. Fig. 3C: Although beta cells can produce serotonin, I am unclear how serum levels reflect the contribution of the islet. The serum will largely reflect the large amount of serotonin from chromaffin cells of the gastrointestinal tract.
6. The kynurenine pathway also leads to NAD synthesis. Insulin secretion is responsive to beta-cell glucose metabolism to ATP. Since NAD is an important co-factor for the TCA cycle are the mice also deficient in NAD?

Reviewer #2 (Remarks to the Author):

Fields et al. report on the development and characterization of a novel model to induce vitamin B6 deficiency in mouse and they study the influence on glucose homeostasis and pancreatic beta cells during pregnancy. In this work, the authors provide some evidence that vitamin B6 deficiency impact glucose metabolism and beta cells proliferation by decreasing serotonin levels. The study is interesting, and it is the first one reporting a possible link between vitamin B6 deficiency and hyperglycemia in pregnancy, an important precondition for the development of gestational diabetes. However, in the current manuscript there are some limitation to the data and the interpretation of the findings. I believe that addressing the comments below will be beneficial to improve the quality of the study.

Major Comments:

- The authors should provide more details about the vitamin B6 deficient diet proposed in this study, in order to ensure reproducibility of the proposed model.
- In the results part at line 209-213 and 232-236 important statements about body weight and food intake are made, however data are not shown and should be included to complete the metabolic phenotype. Alternatively, these statements should be removed.
- In Fig 1A the authors show reduced levels of vitamin B6 at GD12.5, and subsequently perform metabolic phenotype at GD 16.5 (Fig 1B-C). What are the levels of Vitamin B6 at GD16.5? Why this inconsistency among the analysed time points?
- In Fig 1B the CTR group shows fasting glucose levels below 150 mg/dl. However, in the GTT shown in Fig 1C the CTR group does not show the same fasting glucose levels as in Fig 1B and differences in fasting glucose between CTR and vitamin B6 deficient group are lost. Please clarify this discrepancy among the data.
- Differences in glucose clearance are frequently related to defects in insulin secretion. Looking that the GTT curve showed by the vitamin B6 deficiency pregnant group (Fig 1C) it is likely to suppose that insulin secretion is affected and insulin secretion Test (IST) should be performed to support their conclusions.
- In lines 223 -225 the authors state “Consistent to our hypothesis, the studies showed that dietary vitamin B6 deficiency increased fasting glucose levels and induced glucose intolerance during pregnancy.” This sentence should be corrected as following “Consistent to our hypothesis, the studies showed that dietary vitamin B6 deficiency increased fasting glucose levels and exacerbates glucose intolerance during pregnancy.” because pregnancy itself induces “physiological glucose intolerance state”(as shown also by the data from the CTRs in Fig 1C and Fig 2C)
- At line 255-259, the authors state” Our measurement demonstrated that islet levels of serotonin (Fig 3B) correlated positively with its peripheral levels as vitamin B6 deficient dams also had significantly reduced maternal serum serotonin levels compared to controls at GD 16.5 (Fig 3C)”. I find difficult to agree with this statement as they correlate serotonin measurements at different time points (IF at GD12.5 and serum measurements at GD16.5) and different anatomical sites. Further, GD 12.5 and GD16.5 are very different gestational time points with diverse metabolic status and hormonal profiles and therefore not comparable. Data showing serotonin serum level at GD12.5 and IF staining of pancreatic islet at GD16.5 in both experimental groups should be added to support their conclusions.
- In Fig 3C relative serotonin levels are shown and more technical details are required.
- Line 266-267 the author mention that they stained pancreatic islets for MCM-2 to investigate proliferation, however a ki67 or EdU staining would be recommended. Further, when describing IF data on beta cells proliferation they refer to Fig 4A which shows serotonin staining, not indicative of proliferating cells. Please add appropriate IF images or correct mislabeled figure.
- Data presented in Fig 4C should be complemented with graphs of GTT curve and quantification of fasting glucose levels.

Minor Comment:

- At line 26, delete explanation of Greek letter beta.
- It would be useful to indicate precisely and consistently in the results section at which GD the data have been collected.

Reviewer #3 (Remarks to the Author):

The manuscript “B6 deficiency disrupts serotonin signaling in pancreatic islets and induces gestational diabetes in mice,” reports the results of a vitamin B6 deficient diet in gestation. This is a well-written, paper with the novel finding that vitamin B6 deficiency, only in pregnant mice, induces glucose intolerance. This study identifies a novel potential contributor to the pathophysiology of gestational diabetes and is an important addition to the literature. The manuscript does a particularly good job describing the relevant details in the methods section, such that it could be repeated by other researchers. The experimental design is appropriate, with inclusion of relevant controls, and validation of the vitamin B6 deficiency is nicely thorough. The inclusion of the serotonin agonist provides important supporting evidence for the proposed hypothesis. The discussion could do a better job discussing some drawbacks/ alternative interpretations, as detailed below.

The glucose values for all groups (particularly the controls) are really, strikingly higher than are usually reported for pregnant C57Bl/6 mice. Is there something unusual about the meter that was used? Authors should comment on this in the discussion, at least.

A figure, table (or supplemental table) showing the maternal weight and food consumption data would be helpful. Although there were no differences, these are important data.

In figure 4C, it's hard to discern from the overlapping brackets which pairwise comparisons are being made, and both comparisons use the same symbol for significance. Is VB6D + agonist being compared to the control with PBS or the VB6D + PBS? Likewise, in the results text description of this result, line 300, it isn't clear what comparison the $p = 0.0185$ goes with; the preceding text says that VB6D + agonist is not different from control. These need to be clarified.

It seems possible that the B6 deficient diet reduced serotonin production at non-islet sources, thereby reducing systemic, peripheral serotonin, and that led to the lower levels in the islet. Without direct measurement of islet serotonin synthesis or Tph expression/activity only indirect inferences can be made about where the serotonin comes from. This should be acknowledged in the discussion

The paper should discuss how the difference in beta cell proliferation with B6 deficiency and lack of change in beta cell mass can be reconciled. If beta cell mass isn't actually changing, then what is responsible for the loss of glucose tolerance? Could it have more to do with glucose-stimulated insulin release? Is the mass difference only apparent later in gestation?

Reply to the comments on manuscript “Vitamin B6 deficiency disrupts serotonin signaling in pancreatic islets and induces gestational diabetes in mice”, COMMSBIO-20-1322A, Fields *et al.*

Review #1

Comment 1.1: *Did vitamin B6 deficiency affect body weight gestational weight gain or food intake in the mice? Tables should be provided in the paper.*

Reply 1.1: We did not see significant differences in gestational weight gain or food intake in control versus vitamin B6 deficient mice (**Page 12, Paragraph 1**). The data are now included in supplementary materials (**Supplemental Tables 2 and 3**).

Comment 1.2: *The authors claim they are modeling gestational diabetes, but vitamin B6 is present throughout pregnancy. A more detailed characterization of maternal glycemia at earlier timepoints in pregnancy would be appreciated.*

Reply 1.2: We appreciate this suggestion. We added two additional time points (i.e., gestational day 9.5, and 3 weeks postpartum) for further characterization of maternal glycemia and glucose tolerance in our mouse model. Additionally, we have added maternal fasting blood glucose at gestational day 12.5. We therefore now report weight, fasting glucose, glucose AUC, and glucose time graphs for non-pregnant, GD 9.5, GD 12.5, GD 16.5, and 3 weeks postpartum. Our data reveal that vitamin B6 deficiency-induced glucose intolerance develops during mid gestation, starting at GD 9.5 and lasting at least through GD 16.5. The vitamin B6 deficient mice are not glucose intolerant prior to pregnancy and at 3 weeks postpartum, strengthening that it is a model of gestational diabetes. These data have been compiled together in **Figure 1**.

Comment 1.3: *The authors do not show that there are impacts on vitamin B6 deficiency on insulin. How did vitamin B6 affect circulating insulin and insulin following glucose administration (on the GTT)? Functional assays on glucose stimulated insulin secretion by the islets are also necessary to determine the function impact on the islets.*

Reply 1.3: We thank the reviewer for the question and suggestion. To address how vitamin B6 affects circulating insulin and insulin level after glucose administration, we have included new data on glucose stimulated insulin secretion (GSIS), *in vivo*, at both gestational day 12.5 and 16.5. We measured insulin secretion levels at time = 0 (fasting), 15, and 30 minutes post glucose administration (**Figures 2A, C**). Surprisingly, we did not observe differences in insulin secretion as the vitamin B6 deficient mice had normal insulin levels at 0, 15, and 30 minutes post glucose injections. Our results suggest that vitamin B6 deficient mice have normal beta islet function (**Page 13, Paragraph 2**).

Comment 1.4: *Beyond the histological analyses reported in this study, the analyses should be supported by biochemical measurements of insulin and serotonin within the mouse islets.*

Reply 1.4: We have been refining our method for measuring serotonin in the pancreatic islets using liquid chromatography coupled with high resolution mass spectrometry (LC-HRMS). We have included the results for the absolute islet serotonin measurement by LC-HRMS in **Figure 3C**. For biochemical measurements of insulin, we have measured fasting insulin and insulin secretion in the blood plasma at GD 12.5 and 16.5 by ELISA (**Figures 2A and 2C**, respectively).

Comment 1.5: *Figure 3C: Although beta cells can produce serotonin, I am unclear how serum levels reflect the contribution of the islets. The serum will largely reflect the large amount of serotonin from chromaffin cells of the gastrointestinal tract.*

Reply to the comments on manuscript “Vitamin B6 deficiency disrupts serotonin signaling in pancreatic islets and induces gestational diabetes in mice”, COMMSBIO-20-1322A, Fields *et al.*

Reply 1.5: We appreciate the comment. It is true that chromaffin cells contribute to the circulating serotonin levels in mice, however, Kim *et al* (*Nature Medicine* 2010) reports that while islet and serum serotonin levels are elevated during pregnancy, gut serotonin levels are unchanged, so it is unlikely that chromaffin cells in the gut are contributing to the pregnancy-related serotonin levels [1]. We have added LC-MS analysis of maternal serum serotonin at GD 12.5 (**Figure 3D**) and 16.5 (**Figure 3E**).

Comment 1.6: *The kynurenine pathway also leads to NAD synthesis. Insulin secretion is responsive to beta cell glucose metabolism to ATP. Since NAD is an important cofactor for the TCA cycle are these mice also deficient in NAD?*

Reply 1.6: We have measured NAD levels in maternal liver and islets at gestational day 12.5. Interestingly, we see increased NAD in the liver of our vitamin B6 deficient mice, compared to control (**Rebuttal Figure 1 below, left**). We are not the first to see this phenomenon as Bender *et al* saw elevated NAD levels in hepatocytes isolated from vitamin B6 depleted ICR mice [2]. Contradicting to the liver data, the islet has a trend in decreased NAD levels compared to control (**Rebuttal Figure 1, right**); however, we saw normal GSIS *in vivo*, indicating beta cell depolarization is intact. We think that although vitamin B6 is a co-factor in the NAD *de novo* synthesis pathway, there are many factors that can induce NAD production (i.e., inflammation, salvage pathway, etc. as reported in [3, 4]) which could be directly or indirectly be influenced by the deficiency.

Rebuttal Figure 1: **NAD peak area per mg of tissue weight measurements in liver (left) and islets (right).** Using LC-HRMS, relative NAD levels were significantly increased in livers of vitamin B6 deficient dams at GD 12.5 (left), but opposite results were seen in the islets although not statistically significant (right).

Reviewer #2:

Comment 2.1: *The authors should provide more details about the vitamin B6 deficient diet proposed in this study, in order to ensure reproducibility of the proposed model.*

Reply 2.1: We agree with this comment and have added a table in the supplemental section (**Supplementary Table 1**) about the diet composition along with the diet reference number.

Reply to the comments on manuscript “Vitamin B6 deficiency disrupts serotonin signaling in pancreatic islets and induces gestational diabetes in mice”, COMMSBIO-20-1322A, Fields *et al.*

Comment 2.2: *In the results part at line 209-213 and 232-236 important statements about body weight and food intake are made, however data are not shown and should be included to complete the metabolic phenotype. Alternatively, these statements should be removed.*

Reply 2.2: Thank you for this comment and we have included the body weight and gestational food intake data in the supplemental section as **Supplemental Tables 2 and 3**, respectively.

Comment 2.3: *In Fig 1A the authors show reduced levels of vitamin B6 at GD12.5, and subsequently perform metabolic phenotype at GD 16.5 (Fig 1B-C). What are the levels of Vitamin B6 at GD16.5? Why this inconsistency among the analyzed time points?*

Reply 2.3: We appreciate the suggestion for clarifying the inconsistency among time points. We have added data on vitamin B6 levels at GD 16.5. We have also included new data points (including the GD 12.5 metabolic data) so that the analyzed timepoints are consistent.

For clarity purposes, we have moved results on reduced levels of vitamin B6 to the supplemental section (**Supplemental Figure 5A**). We also performed the same measurements at GD 16.5 to draw more direct comparison with glucose tolerance timepoints (**Supplemental Figure 5B**). Throughout the revised manuscript, we cautiously emphasized which timepoints were being experimentally tested and referenced.

Comment 2.4: *In Fig 1B the CTR group shows fasting glucose levels below 150 mg/dl. However, in the GTT shown in Fig 1C the CTR group does not show the same fasting glucose levels as in Fig 1B and differences in fasting glucose between CTR and vitamin B6 deficient group are lost. Please clarify this discrepancy among the data.*

Reply 2.4: We appreciate your close attention to details for our figures. Please note that in this revised manuscript, we have changed the layout of **Figure 1**. **Figure 1B** now shows the fasting glucose at GD 16.5 and **Figure 1E** the corresponding GTT glucose time graphs. For analyses of fasting glucose and glucose time graphs, we have used different statistical methods. For **Figure 1B** (analysis of fasting glucose), we used an unpaired, two-sided, t-test. For **Figure 1E** (analysis of glucose time graphs), we conducted a two-way ANOVA with repeated measures with Tukey’s *post hoc analysis*, which adjusted the p-value for multiple comparisons. When multiple comparisons were done, we did not see statistical significance in the fasting glucose between the two groups. If we analyze the data using Fisher exact LSD (ignoring multiple comparisons), for example, we do see statistical significance in the fasting glucose (at T=0 minutes). Lastly, we have adjusted the graph axes so that they are now equivalent when reporting fasting glucose (**Figure 1B**) and glucose time graphs (**Figures 1C-E**). Please note that the fasting glucose (**Figure 1B**) are the same as T=0 for the glucose time graphs (**Figures 1C-E**, and **Supplemental Figures 6A, B**)

Comment 2.5: *Differences in glucose clearance are frequently related to defects in insulin secretion. Looking that the GTT curve showed by the vitamin B6 deficiency pregnant group (Fig 1C) it is likely to suppose that insulin secretion is affected and insulin secretion Test (IST) should be performed to support their conclusions.*

Reply 2.5: Thank you this comment. We have performed GSIS at both gestational day 12.5 and 16.5. The results of these experiment are presented in **Figures 2A and 2C**. We did not observe hypo- or hyper-insulin secretion during GSIS, indicating that insulin secretion is not different relative to controls. We reported the findings in Results section (**Page 13, Paragraph 2**).

Reply to the comments on manuscript “Vitamin B6 deficiency disrupts serotonin signaling in pancreatic islets and induces gestational diabetes in mice”, COMMSBIO-20-1322A, Fields *et al.*

Comment 2.6: *In lines 223-225 the authors state “Consistent to our hypothesis, the studies showed that dietary vitamin B6 deficiency increased fasting glucose levels and induced glucose intolerance during pregnancy.” This sentence should be corrected as following “Consistent to our hypothesis, the studies showed that dietary vitamin B6 deficiency increased fasting glucose levels and exacerbates glucose intolerance during pregnancy.” because pregnancy itself induces “physiological glucose intolerance state” (as shown also by the data from the CTRs in Fig 1C and Fig 2C).*

Reply 2.6: Thank you for carefully reviewing the phrasing of our results, we appreciate the revised sentence suggestion. With the revisions and additions of new data, we have incorporated the suggested phrase with some modifications in the Discussion. We have phrased it as “Although pregnancy itself induces a “physiological state of glucose intolerance”, our studies show that vitamin B6 deficiency exacerbates gestational glucose intolerance”. (**Page 18, Paragraph 1**).

Comment 2.7: *At line 255-259, the authors state” Our measurement demonstrated that islet levels of serotonin (Fig 3B) correlated positively with its peripheral levels as vitamin B6 deficient dams also had significantly reduced maternal serum serotonin levels compared to controls at GD 16.5 (Fig 3C)”. I find difficult to agree with this statement as they correlate serotonin measurements at different time points (IF at GD12.5 and serum measurements at GD16.5) and different anatomical sites. Further, GD 12.5 and GD16.5 are very different gestational time points with diverse metabolic status and hormonal profiles and therefore not comparable. Data showing serotonin serum level at GD12.5 and IF staining of pancreatic islet at GD16.5 in both experimental groups should be added to support their conclusions.*

Reply 2.7: Thank you for your comment about our overstatement between GD 12.5 islet serotonin and GD 16.5 serum serotonin. We agree that we need to compare islet and serum serotonin at the same timepoints. We have included new data for GD 12.5 which includes islet serotonin levels (**Figure 3C**) and serum serotonin levels (**Figure 3D**) from control and vitamin B6 deficient dams. Based on our new results, at GD 12.5, only islet serotonin is reduced in vitamin B6 deficient dams while serum serotonin is unchanged. We reported these findings in the results (**Page 14, Paragraph 2; Page 15, Paragraph 2**) and included them in the discussion (**Page 18**). We have also included additional pancreatic staining data for GD16.5 (**Supplemental Figure 8F; Pages 15 and 16 last and first paragraphs**, respectively).

Comment 2.8: *In Fig 3C relative serotonin levels are shown and more technical details are required.*

Reply 2.8: Thank you for suggesting that we need more technical details pertaining to our method. We have included technical details of the method (**Pages 8 and 9**) and have added the statement “Peak areas corresponding to levels of endogenous serotonin or the spiked internal serotonin standard were calculated using the LC Quan node of the XCalibur software (Thermo Fisher). To determine the relative levels of serum serotonin, we calculated the ratio between endogenous serotonin and the internal serotonin standard” (**Page 9, Paragraph 2**). Please note that in this resubmission, the LC-MS relative serotonin levels measurements are presented in **Figures 3D and 3E** (for maternal serum at GD 12.5 and 16.5, respectively).

Comment 2.9: *Line 266-267 the author mention that they stained pancreatic islets for MCM-2 to investigate proliferation, however a ki-67 or EdU staining would be recommended. Further,*

Reply to the comments on manuscript “Vitamin B6 deficiency disrupts serotonin signaling in pancreatic islets and induces gestational diabetes in mice”, COMMSBIO-20-1322A, Fields *et al.*

when describing IF data on beta cells proliferation they refer to Fig 4A which shows serotonin staining, not indicative of proliferating cells. Please add appropriate IF images or correct mislabeled figure.

Reply 2.9: We appreciate your comment and suggestion for proliferation markers, along with identifying the mislabeled figure. We have corrected the **Figure 4A** by fixing the label to MCM-2. Our intention was to use ki-67 because it is well-referenced and covers all stages of the cell cycle. However, our challenge was that the insulin antibody was raised in rabbit, similar to most ki-67 antibodies on the market. There were only two ki-67 antibodies raised in other hosts that react with mouse, the rat anti-Ki-67 monoclonal (eBioscience™, Invitrogen, Clone SolA15; 1:100 – 1:750) and mouse anti-Ki-67 monoclonal (BD Pharmingen™, Clone B56, 1:100 – 1:750); these antibodies were tested but did not work. Additionally, we could not use EdU or BrdU staining because the mice were already euthanized, and pancreases were isolated and fixed. Although there is not much literature using MCM-2 for beta cell proliferation, there is growing use of MCM-2 in general especially in the cancer field [5]. Additionally, our control GD 12.5 dams had similar beta cell proliferation (~10 %) with Kim, K *et al.* (~15%), which utilized BrdU [1]. Furthermore, our vitamin B6 deficient mice had similar proliferation (~5%) as the *Htr2b* knockout mice (~6%) [1]. We are confident that MCM-2 is a reliable proliferation marker for quantifying beta cell proliferation.

Comment 2.10: *Data presented in Fig 4C should be complemented with graphs of GTT curve and quantification of fasting glucose levels.*

Reply 2.10: Thank you for that suggestion. We have added the fasting glucose levels (**Supplemental Figure 10B**) along with the GTT time graphs (**Supplemental Figure 10C**) to the manuscript and added a discussion statements addressing the possibility of repeated injections influence a stress response in our mice (**Page 20, Paragraph 1**).

Comment 2.11: *At line 26, delete explanation of Greek letter beta.*

Reply 2.11: We have deleted “beta” and changed all “beta” to the symbol “β” for consistency.

Comment 2.12: *It would be useful to indicate precisely and consistently in the results section at which GD the data have been collected.*

Reply 2.12: Thank you for expressing this comment and we have revised our manuscript to ensure the reader knows precisely what time point is being addressed.

Reviewer #3:

Comment 3.1: *The glucose values for all groups (particularly the controls) are really, strikingly higher than are usually reported for pregnant C57Bl/6 mice. Is there something unusual about the meter that was used? Authors should comment on this in the discussion, at least.*

Reply 3.1: For all our studies, we use the AlphaTrak® 2 blood glucometer which has been shown to be appropriate for measuring blood glucose in mice, specifically C57BL/6J strain [6]. We have searched the literature comparing our fasting glucose levels to other studies. We have found some studies use enzymatic assays which measure the amount of plasma glucose [7, 8], while handheld glucometers measure glucose from whole blood [9-11]. Additionally, many laboratories use human glucometers [9-11], while our studies used AlphaTRAK glucometer which is intended for animal use. These differences alone can contribute to differences in fasting blood glucose seen in

Reply to the comments on manuscript “Vitamin B6 deficiency disrupts serotonin signaling in pancreatic islets and induces gestational diabetes in mice”, COMMSBIO-20-1322A, Fields *et al.*

the literature. Thus, it is important to specify the exact enzymatic kit or handheld glucometer used in the study. A 2010 study demonstrates that there are differences between enzymatic assays and glucometers for glucose measurements [6]. Further in this study, the authors compared glucose measurement between AlphaTRAK monitor and enzymatic assay using the same samples [6]. The linear regression equation from this study (please see **Rebuttal Figure 2** below) can be used to estimate enzymatic assay plasma glucose values (mM; column 2 in the table below) from the AlphaTRAK glucose values (mg/dL; column 1 in the table below). Additionally, the third column converts the estimated plasma glucose values from mM to mg/dL plasma glucose values for same unit comparison between the AlphaTRAK glucometer and enzymatic assay plasma glucose values. The data do show that the AlphaTRAK glucometer produces *higher* glucose values, but they are still in line with normal mouse physiology plasma glucose [6], compared to the enzymatic plasma glucose assay.

Our fasting conditions were for 6 hours from 7:00am to 1:00pm, which is common to produce fasting glucose levels that are 100 to 160 mg/dL [12]. Overnight fasting typical produces lower fast glucose levels in mice < 100 mg/dL [12], but it is not advised for pregnant mice. Lastly, we always have control and vitamin B6 deficient mice represented on the day of experiments and compare the vitamin B6 deficient dams to their respective controls.

Equation of regression line: $y=0.0360x + 1.8610$		
AlphaTRAK (mg/dL)	Enzymatic Assay: plasma glucose (mM)	Converted: plasma glucose (mg/dL)
100	5.461	98.3973902
150	7.261	130.8301502
180	8.341	150.2898062
200	9.061	163.2629102
250	10.861	195.6956702

Rebuttal Figure 2: Graph from [6] (left) and table data interpolated from the regression line (right). The graph on the left shows the correlation between AlphaTRAK monitor glucose measurement from whole blood (x-axis) compared to plasma glucose from an enzymatic assay taken from the same blood sample (y-axis) and was analyzed by linear regression. On the right, we created a table, from the linear regression equation, to show the direct comparison between the AlphaTRAK glucometer glucose values and plasma glucose from enzymatic assay.

Comment 3.2: *A figure, table (or supplemental table) showing the maternal weight and food consumption data would be helpful. Although there were no differences, these are important data.*

Reply to the comments on manuscript “Vitamin B6 deficiency disrupts serotonin signaling in pancreatic islets and induces gestational diabetes in mice”, COMMSBIO-20-1322A, Fields *et al.*

Reply 3.2: We have included these data in the supplemental section as **Supplemental Tables 2 and 3**.

Comment 3.3: *In figure 4C, it’s hard to discern from the overlapping brackets which pairwise comparisons are being made, and both comparisons use the same symbol for significance. Is VB6D + agonist being compared to the control with PBS or the VB6D + PBS? Likewise, in the results text description of this result, line 300, it isn’t clear what comparison the $p = 0.0185$ goes with; the preceding text says that VB6D + agonist is not different from control. These need to be clarified.*

Reply 3.3: We apologize for not clearly defining the pairwise comparisons and we have modified the graphs and texts to clearly represent the different pairwise comparisons (**Page 16, Last Paragraph – Page 17, Paragraph 1**). In the PBS group, we compared control and vitamin B6 deficient mice to show that the vitamin B6 deficient dams were glucose intolerant ($p = 0.0313$, the PBS group in **Figure 4C**). Next, we compared the impact of agonist treatment in our vitamin B6 deficient mice; so here we compared VB6D + PBS vs VB6D + Agonist (grey bars in **Figure 4C**) to show that treatment with the HTR2B improved the glucose intolerance phenotype ($p = 0.0185$). Our text has been modified to indicate this information so that it is clear which pairwise comparisons are made (**Page 16, Last Paragraph – Page 17, Paragraph 1**). We tested if there was a diet x treatment interaction using a two-way ANOVA and indeed observed an interaction effect and proceeded to *post hoc* pairwise analyses.

Comment 3.4: *It seems possible that the B6 deficient diet reduced serotonin production at non-islet sources, thereby reducing systemic, peripheral serotonin, and that led to the lower levels in the islet. Without direct measurement of islet serotonin synthesis or Tph expression/activity only indirect inferences can be made about where the serotonin comes from. This should be acknowledged in the discussion.*

Reply 3.4: Thank you for your comment and we understand that we are systemically reducing vitamin B6 levels and this will affect all areas of serotonin production. We agree that without measuring the enzymes in the tryptophan-serotonin catabolism pathway within the islets, it is unclear whether the reduced vitamin B6 levels is directly or indirectly causing reduced islet serotonin levels. Our LC-HRMS analysis shows reduction in islet serotonin levels in the vitamin B6 deficient dams at GD12.5, but systemically, lower serum serotonin was only observed at GD16.5, suggesting that vitamin B6 deficiency impacts the islets prior to impacting systemic serotonin levels. We have added a paragraph mentioning the islet specific effects at GD 12.5 and acknowledging that islet *TPHI* expression or activity was not measured (**Page 18, Paragraph 1**). In lieu of *TPHI* measurement, we undertook a different approach of targeted LC-HRMS to measure islet serotonin, tryptophan and pyridoxal 5’ phosphate (PLP) (**Supplemental Figure 7**) to provide insights into how the tryptophan catabolism and serotonin levels were altered in the pancreatic islets. We did not observe impaired tryptophan catabolism (i.e., we did not observe elevated levels of tryptophan in the vitamin B6 deficient mice; **Supplemental Figure 7A**) suggesting that the *TPHI* activity is not impaired. Because we observed reduced islet PLP (**Supplemental Figure 7B**), we think that one explanation could be that the pathway downstream of *TPHI* might be affected (e.g., the conversion of 5 hydroxytryptophan into serotonin which is PLP-dependent as shown in **Supplemental Figure 2**). We have included this statement in Discussion (**Page 18, Paragraph 1 – Page 19, Paragraph 1**)

Reply to the comments on manuscript “Vitamin B6 deficiency disrupts serotonin signaling in pancreatic islets and induces gestational diabetes in mice”, COMMSBIO-20-1322A, Fields *et al.*

Comment 3.5: *The paper should discuss how the difference in beta cell proliferation with B6 deficiency and lack of change in beta cell mass can be reconciled. If beta cell mass isn't actually changing, then what is responsible for the loss of glucose tolerance? Could it have more to do with glucose-stimulated insulin release? Is the mass difference only apparent later in gestation?*

Reply 3.5: We understand and appreciate your comment. We have performed the additional experiments to characterize our mouse model. We looked at beta cell mass later in pregnancy, at GD 16.5, and saw a trend in reduced beta cell mass ($p = 0.0726$; **Supplemental Figure 8F**). We also performed glucose stimulated insulin secretion at GD 12.5 and 16.5, and did not see any significant differences between treatment groups at any time point, suggesting that the residual beta cells were able to secrete sufficient insulin. We also performed insulin tolerance testing on our mice at GD 12.5 and 16.5. At GD 12.5, we observed vitamin B6 deficient dams were insulin resistant (**Figure 2A**), while our GD 16.5 data showed that the vitamin B6 deficient dams had higher insulin sensitivity compared to controls (**Figure 2C**). Overall, our data suggest that the mechanisms are complex, and that the glucose intolerance is caused by insulin resistance and other insulin-independent mechanisms. In our discussion section, we discussed other potential mechanisms including expression and activity of various glucose transporter (e.g., GLUT1 and GLUT4), glucose effectiveness, and counterregulatory mechanisms (**Page 19, Paragraph 2**).

References:

1. Kim, H., et al., *Serotonin regulates pancreatic beta cell mass during pregnancy*. Nat Med, 2010. **16**(7): p. 804-8.
2. Bender, D.A., E.N. Njagi, and P.S. Danielian, *Tryptophan metabolism in vitamin B6-deficient mice*. Br J Nutr, 1990. **63**(1): p. 27-36.
3. Osterman, A., *Biogenesis and Homeostasis of Nicotinamide Adenine Dinucleotide Cofactor*. EcoSal Plus, 2009. **3**(2).
4. Xie, N., et al., *NAD(+) metabolism: pathophysiologic mechanisms and therapeutic potential*. Signal Transduct Target Ther, 2020. **5**(1): p. 227.
5. Jurikova, M., et al., *Ki67, PCNA, and MCM proteins: Markers of proliferation in the diagnosis of breast cancer*. Acta Histochem, 2016. **118**(5): p. 544-52.
6. Ayala, J.E., et al., *Standard operating procedures for describing and performing metabolic tests of glucose homeostasis in mice*. Dis Model Mech, 2010. **3**(9-10): p. 525-34.
7. Plows, J.F., et al., *Absence of a gestational diabetes phenotype in the LepRdb/+ mouse is independent of control strain, diet, misty allele, or parity*. Sci Rep, 2017. **7**: p. 45130.
8. Yamashita, H., et al., *Effect of spontaneous gestational diabetes on fetal and postnatal hepatic insulin resistance in Lepr(db/+) mice*. Pediatr Res, 2003. **53**(3): p. 411-8.
9. Yao, L., et al., *Resveratrol relieves gestational diabetes mellitus in mice through activating AMPK*. Reprod Biol Endocrinol, 2015. **13**: p. 118.
10. Banerjee, R.R., et al., *Gestational Diabetes Mellitus From Inactivation of Prolactin Receptor and MafB in Islet beta-Cells*. Diabetes, 2016. **65**(8): p. 2331-41.
11. Susiarjo, M., et al., *Bile acids and tryptophan metabolism are novel pathways involved in metabolic abnormalities in BPA-exposed pregnant mice and male offspring*. Endocrinology, 2017.

Reply to the comments on manuscript “Vitamin B6 deficiency disrupts serotonin signaling in pancreatic islets and induces gestational diabetes in mice”, COMMSBIO-20-1322A, Fields *et al.*

12. Benede-Ubieto, R., et al., *Guidelines and Considerations for Metabolic Tolerance Tests in Mice*. *Diabetes Metab Syndr Obes*, 2020. **13**: p. 439-450.

REVIEWERS' COMMENTS:

Reviewer #1 (Remarks to the Author):

The authors have provided new data and considerably revised the manuscript to appropriately address the previous review comments. The authors report that vitamin B6 deficiency induces glucose intolerance at GD12.5 and GD16.5. Interestingly, the authors show that vitamin B6 deficiency did not impact glucose-stimulated insulin secretion but did induce insulin resistance. This suggests that rather than at the level of the islet, vitamin B6 deficiency dysregulates the PLP-dependent transaminases and glycogen phosphorylase enzymes that have an established role in gluconeogenesis in the liver and mediate impaired glucose tolerance, which is a well-established phenomenon (see Combs GF 2007 *The Vitamins: Fundamental Aspects in Nutrition and Health*).

Reviewer #2 (Remarks to the Author):

After careful assessment of the revised manuscript, I think the study presented by Fields et al., remarkably improved. I am pretty pleased with the additional experiments and the clarifications provided in response to the reviewer's comments. Also, the authors well harmonized the experimental suggestions from the reviews in the new manuscript layout. Just one minor comment, please add in Suppl table 2 and Suppl table 3 the unit of the reported values.

Reviewer #3 (Remarks to the Author):

All of my initial comments were addressed. The mouse model presented, as well as the careful demonstration that gestational B6 deficiency causes glucose intolerance, are valuable additions to the literature. The fuller data presentation from each of the timepoints and the additional early and post-partum timepoints add considerably to the manuscript.

The conclusion of the paper, at least as stated in the abstract, should be changed in light of the new data showing that insulin secretion is not affected by the vitamin B6 deficiency. From the abstract "...These findings demonstrate that vitamin B6 deficiency influences maternal glucose tolerance in a serotonin-dependent mechanism." While the experiments do show that B6 deficiency reduces islet serotonin, beta cell proliferation and glucose tolerance, they do not show that serotonin loss is the mechanism by which B6 deficiency influences glucose tolerance. Since islet serotonin loss is generally understood to cause glucose intolerance by reducing insulin production (directly and indirectly), and this doesn't seem to be what's happening, the mechanism by which the gestational B6 deficiency causes glucose intolerance has not yet been elucidated. This is handled appropriately in the discussion, but the abstract should better reflect the more nuanced conclusion.